# Homozygosity for a Novel *DOCK7* Variant Due to Segmental Uniparental Isodisomy of Chromosome 1 Associated with Early Infantile Epileptic Encephalopathy (EIEE) and Cortical Visual Impairment

**DOI:** 10.3390/ijms23137382

**Published:** 2022-07-02

**Authors:** Fatma Kivrak Pfiffner, Samuel Koller, Anika Ménétrey, Urs Graf, Luzy Bähr, Alessandro Maspoli, Annette Hackenberg, Raimund Kottke, Christina Gerth-Kahlert, Wolfgang Berger

**Affiliations:** 1Institute of Medical Molecular Genetics, University of Zurich, Wagistrasse 12, 8952 Schlieren, Switzerland; kivrakpfiffner@medmolgen.uzh.ch (F.K.P.); koller@medmolgen.uzh.ch (S.K.); graf@medmolgen.uzh.ch (U.G.); baehr@medmolgen.uzh.ch (L.B.); maspoli@medmolgen.uzh.ch (A.M.); 2Department of Pediatric Neurology, University Children’s Hospital, University of Zurich, 8032 Zurich, Switzerland; anika.menetrey@kispi.uzh.ch (A.M.); annette.hackenberg@kispi.uzh.ch (A.H.); 3Department of Diagnostic Imaging, University Children’s Hospital, University of Zurich, 8032 Zurich, Switzerland; raimund.kottke@kispi.uzh.ch; 4Department of Ophthalmology, University Hospital, 8091 Zurich, Switzerland; christina.gerth-kahlert@usz.ch; 5Neuroscience Center Zurich, University and ETH Zurich, 8057 Zurich, Switzerland; 6Center for Integrative Human Physiology, University of Zurich, 8057 Zurich, Switzerland

**Keywords:** uniparental disomy (UPD), mUPiD, UPiD, UPhD, loss of heterozygosity (LOH), early infantile epileptic encephalopathy 23, EIEE23, cortical blindness, cortical visual impairment

## Abstract

Early infantile epileptic encephalopathy (EIEE) is a severe neurologic and neurodevelopmental disease that manifests in the first year of life. It shows a high degree of genetic heterogeneity, but the genetic origin is only identified in half of the cases. We report the case of a female child initially diagnosed with Leber congenital amaurosis (LCA), an early-onset retinal dystrophy due to photoreceptor cell degeneration in the retina. The first examination at 9 months of age revealed no reaction to light or objects and showed wandering eye movements. Ophthalmological examination did not show any ocular abnormalities. The patient displayed mildly dysmorphic features and a global developmental delay. Brain MRI demonstrated pontine hypo-/dysplasia. The patient developed myoclonic epileptic seizures and epileptic spasms with focal and generalized epileptiform discharges on electroencephalogram (EEG) at the age of 16 months. Genetic screening for a potentially pathogenic DNA sequence variant by whole-exome sequencing (WES) revealed a novel, conserved, homozygous frameshift variant (c.5391delA, p.(Ala1798Leu*fs*Ter59)) in exon 42 of the *DOCK7* gene (NM_001271999.1). Further analysis by SNP array (Karyomapping) showed loss of heterozygosity (LOH) in four segments of chromosome 1. WES data of the parents and the index patient (trio analysis) demonstrated that chromosome 1 was exclusively inherited from the mother. Four LOH segments of chromosome 1 alternately showed isodisomy (UPiD) and heterodisomy (UPhD). In WES data, the father was a noncarrier, and the mother was heterozygous for this *DOCK7* variant. The *DOCK7* gene is located in 1p31.3, a region situated in one of the four isodisomic segments of chromosome 1, explaining the homozygosity seen in the affected child. Finally, Sanger sequencing confirmed maternal UPiD for the *DOCK7* variant. Homozygous or compound heterozygous pathogenic variants in the *DOCK7* (dedicator of cytokinesis 7) gene are associated with autosomal recessive, early infantile epileptic encephalopathy 23 (EIEE23; OMIM #615,859), a rare and heterogeneous group of neurodevelopmental disorders diagnosed during early childhood. To our knowledge, this is the first report of segmental uniparental iso- and heterodisomy of chromosome 1, leading to homozygosity of the *DOCK7* frameshift variant in the affected patient.

## 1. Introduction

Early infantile epileptic encephalopathy syndrome (EIEE) is a rare and severe neurodevelopmental disorder with high premature mortality characterized by abundant epileptiform activity on electroencephalogram (EEG) and frequent seizures with adverse impact on development in neonates or infants [1]. It is characterized by genetic heterogeneity. One particular subtype (EIEE23, OMIM 615,859) is associated with autosomal recessive mutations in the *DOCK7* gene on human chromosome 1.

The *DOCK7* gene consists of 50 exons and codes for a guanine nucleotide exchange factor (GEF) that plays an essential role in axon formation and neuronal polarization and is highly expressed in the brain during the early stages of development [2,3]. *DOCK7*, together with *TACC3*, controls interkinetic nuclear migration (INM) and neurogenesis of radial glial progenitor cells during cortical development [3]. Pathogenic variants in the *DOCK7* gene were first reported in 2014 in three girls from two nonconsanguineous families presenting with dysmorphic features, early-onset epilepsy, intellectual disability, and cortical blindness, followed by further additional reports [4,5,6,7,8]. Most of the pathogenic variants in *DOCK7* reported so far are truncating variants [4,5,6,7,8]. In addition, one homozygous intragenic tandem duplication was recently shown to trigger nonsense-mediated mRNA decay (NMD) in the patient’s fibroblasts [9].

Uniparental disomy (UPD) leading to autosomal recessive (AR) diseases is a rare phenomenon. In UPD, both copies of a chromosome are derived from a single parent due to errors in meiotic or mitotic cell division. Two major types of UPD can be distinguished: (1) uniparental heterodisomy (UPhD), where both homologs of a pair of chromosomes (grandparental) are transmitted from one parent and (2) uniparental isodisomy (UPiD), where two identical copies of a single chromosome are inherited from one parent [10]. Most UPDs are caused by recombinational events during meiosis giving rise to both heterodisomic and isodisomic segments rather than affecting the whole chromosome [10,11]. While both UPhD and UPiD can result in imprinting disorders, only UPiD may lead to homozygosity of pathogenic sequence variants [11,12]. Since the first demonstration of UPiD in a patient with cystic fibrosis [11,13,14,15,16,17,18,19,20,21], an increasing number of incidental findings of autosomal recessive diseases caused by UPiD have been documented.

In clinical practice, UPD is typically investigated using short tandem repeat (STR) markers, analysis of methylation patterns for the chromosomal region of interest, and/or testing for chromosomal abnormalities using single-nucleotide polymorphism (SNP) arrays [22,23]. In array-based SNP genotyping, a large number of informative SNPs covering the whole genome are used. This technique can detect copy number abnormalities and facilitates the detection of UPD, because homozygosity and heterozygosity can be precisely mapped along the chromosomes using SNPs [24,25]. Furthermore, SNP arrays allow a rapid, cost-effective, and reliable approach. Isodisomic regions are easily identified based on stretches of DNA with loss of heterozygosity (LOH). However, UPhD can only be detected if parental DNA is analyzed [23,25]. Routine SNP arrays cannot determine the parental origin of UPiD/UPhD without comparing patients’ genotypes with the parents’ genotypes by trio-based NGS as well as by SNP array analysis [12,22].

In this study, we report a novel homozygous *DOCK7* sequence variant arising from maternal isodisomy of chromosome 1 in a child with a specific subtype of EIEE.

## 2. Results

### 2.1. Clinical History and Findings

The proband was a 9-month-old girl of a non-consanguineous couple of Caucasian background. Family history was unremarkable. The proband was born at term by cesarean section after an uncomplicated pregnancy, with a normal birthweight of 3290 g. Developmental abnormalities were first noted by her mother at the age of 5 months. The patient was referred to the Department of Ophthalmology for further evaluation of her reduced visual responses at the age of 9 months. The mother reported an improved visual behavior with intermittent fixation of objects. She presented with wandering eye movements, absent fixation to light or objects, and slow pupillary response to light. No abnormalities were noted on the slit lamp and dilated fundus examination. Retinoscopy did not reveal an abnormal refractive error. Based on these findings, a differential diagnosis of Leber congenital amaurosis (LCA) or cerebral disease was made, and further clinical examination was initiated. Scotopic and photopic ERG responses were recordable in the right eye (the left eye was not tested due to restlessness) at the age of 15 months, a finding likely not compatible with LCA. At her last examination at the age of 2 years, the patient fixated light in a darkened room, but not under normal room illumination.

The first neurological examination was performed at the age of 10 months. The girl explored objects orally and by tacti and was able to sit supported with pronounced muscular hypotonia. Over the next months, she made steady developmental progress and was able to sit unsupported at the age of 19 months. At that time, she still showed pronounced motor stereotypies and was able to move while lying on her back by pushing herself forward with her legs. She was not able to crawl or stand. Her speech as well as her cognitive development were severely impaired.

At the age of 16 months, the EEG showed frequent multifocal and generalized sharp slow waves associated with epileptic spasms on simultaneous video recording (Figure 1). Early epileptic encephalopathy was diagnosed. Treatment with valproic acid was not effective. An add-on therapy with vigabatrine was suggested but rejected by the parents. The girl still has daily seizures at the time of writing.

Brain MRI at the age of 10 months showed a dysplastic brainstem with a hypoplastic pons, most prominent in the midline, dilated pontine perivascular spaces anteriorly, and a concave posterior border of the pons. An abnormally marked sulcus at the pontomedullary junction was seen as previously described [4] (Figure 1). The occipital white matter in the region of the visual cortex showed increased high T2 signal compatible with a lower degree of myelination. In addition, diffuse cerebellar microcystic changes and severely hypoplastic olfactory bulbs were observed, findings not reported elsewhere to date.

Eye examination of the mother revealed normal findings, with no features of retinal dystrophy. The father of the proband denied any visual problems but did not receive an eye examination. The remaining medical and family histories were unremarkable.

### 2.2. Molecular and Cytogenetic Analyses

The trio analysis by WES identified a novel, homozygous frameshift variant (c.5391delA, p.(Ala1798Leu*fs*Ter59)) in exon 42 (out of 50 exons) of the *DOCK7* gene (NM_001271999.1) as the most likely causative variant in the proband. In addition, her mother was heterozygous, and her father was a non-carrier of this frameshift variant. Sanger sequencing confirmed these WES data (Figure 2A,B). This frameshift variant results in a premature stop codon (PMC) in the catalytic DHR2 domain (Figure 3A) and most likely leads to the premature decay (NMD) of the corresponding mRNA. According to ACMG guidelines [26], the novel frameshift variant identified in this patient was considered to be pathogenic, based on PVS1 (pathogenic very strong; i.e., a null-variant (frameshift)), PM2 (pathogenic supporting; i.e., not found in gnomAD database), PP3 (pathogenic supporting; i.e., it lies in a highly conserved amino acid position (Figure 3B)), PP4 (pathogenic supporting; patient’s phenotype highly specific for EIEE23), and PP5 (pathogenic supporting; ClinVar classifies the variant as pathogenic).

SNP array analysis showed alternately four stretches of isodisomic and heterodisomic segments of chromosome 1. Four isodisomic segments were visible on the SNP array as loss of heterozygosity (LOH) regions (extensions) at/in 1p36.33p36.21, 1p35.1p31.1, 1p13.1q24.2, and 1q41q43 (shown in blue in Figure 4A,B). The LOH block at 1p35.1p31.1, spanning 47.5 Mb, includes the entire *DOCK7* gene (Figure 4B). SNP haplotype analysis of chromosome 1 revealed that both isodisomic and heterodisomic segments of the patient were derived from the mother and did not involve any paternal origin (Appendix A). Aneuploidy screening by Veriseq showed that the copy number (log R signal intensity) was normal (Figure 4C). Based on the WES dataset of the patient, no CNV was detected in chromosome 1.

To investigate the possibility of an altered imprinted gene dosage as a consequence of maternal uniparental disomy, we checked the imprinted genes in chromosome 1 on the “Geneimprint” portal (https://www.geneimprint.com/site/genes-by-species (accessed on 21 December 2021)). We found that 18 genes predicted to play a role in imprinting were listed in chromosome 1. However, the dosage alteration of these genes could not be related to the disease phenotype.

### 2.3. Investigation of Single-Nucleotide Variants (SNVs) within Chromosome 1

To investigate both mUPDs seen in the SNP array analysis as well as the non-Mendelian inheritance of the *DOCK7* frameshift variant, rare sequence variants on chromosome 1 from the WES data in the index patient were subdivided in both isodisomic (homozygous) and heterodisomic (heterozygous) regions according to their genomic locations (hg19) displayed by the SNP array. A total of 27 rare sequence variants were identified in the isodisomic regions. Of these 27 variants, 20 were homozygous and were inherited only from the heterozygous mother. The remaining seven heterozygous variants were considered to be false positives. A total of 60 heterozygous variants were identified in the heterodisomic regions. Of these, 41 were inherited exclusively from the heterozygous mother, and 19 were found in the father as well. These data are consistent with the results of the SNP array analysis.

Two heterozygous *ABCA4* (NM_000350.2) variants (c.157G>A (p.(Glu53Lys)) and c.2267C>T (p.(Ser756Phe); rs372508062) on chromosome 1 were additionally identified and classified as variants of unknown significance (VUS) according to ACMG guidelines. The unaffected mother carried these two heterozygous variants, whereas the father carried the reference alleles (c.157G and c.2267C). Mutations in the *ABCA4* gene are associated with different types of retinal degeneration, including Stargardt disease (STGD). However, ophthalmic evaluation of the proband and her mother did not reveal any characteristics of STGD. Since these two *ABCA4* variants are located in the maternally inherited heterodisomic region of chromosome 1 of the proband, additional segregation analysis by Sanger sequencing was performed for the maternal grandparents, in order to determine whether these variants are on the same or on different alleles. The maternal grandparents were heterozygous carriers of each *ABCA4* sequence variant, thus both *ABCA4* variants are biallelic in the patient and her unaffected mother (Appendix A).

## 3. Discussion

We report the first case of EIEE23 as a result of segmental maternal UPiD. The patient was homozygous for a novel, potentially pathogenic frameshift variant c.5391delA, (p.(Ala1798Leu*fs*Ter59)) in the *DOCK7* gene (NM_001271999.1). The unusual maternal inheritance was related to the segmental maternal UPiD of chromosome 1. The variant was not present in the Human 1000 Genome Project, HGMD, or gnomAD databases, but was documented in ClinVar (https://preview.ncbi.nlm.nih.gov/clinvar/variation/870658/ (accessed on 21 December 2021)). We considered this frameshift variant as pathogenic and to be associated with EIEE23 due to a strong genotype–phenotype correlation. From the clinical viewpoint, the patient presented with typical characteristics of EIEE23 such as lack of reaction to visual stimuli despite a normal anterior and posterior eye examination, pontine hypoplasia, myoclonic epileptic seizures with focal epileptiform discharges on EEG, and mildly dysmorphic features. To date, eight truncating and two canonical splice variants in the *DOCK7* gene have been described in nine patients as being associated with EIEE23 [4,5,6,7,8,9]. Most of these variants were detected in patients with an age of 3–10 years [4,5,6,7]. These variants caused a severe neurodevelopmental condition characterized by impaired language processing, psychomotor delay, cortical blindness, and infantile spasms. A recently identified homozygous tandem duplication in *DOCK7* (NM_001271999.1: c.390_3936dup) comprising exons 5–31 in an adult sibling pair (23 and 27 years) has been shown to cause *DOCK7* deficiency via NMD in the patients’ fibroblasts [9]. In contrast to the previous cases with biallelic truncating variants, these adult patients developed better language and social communication skills compared to previously published cases. Typical facial features were common in all of the affected patients, including periorbital fullness, low anterior hairline, broad nasal tip, long eyelashes, and telecanthus. Our patient showed similar mildly dysmorphic features such as long eyelashes, broad nose tip, and telecanthus.

DOCK7, a member of the DOCK180-related protein family, is expressed in all major brain regions [2]. It encodes a guanine nucleotide exchange factor (GEF) that activates the Rho family GTPases Rac and/or Cdc42 through DOCK homology region 2 (DHR-2) and is involved in axon development [27]. In vitro experiments demonstrated that DOCK7 regulates microtubule stability in the nascent axon through Rac activation and inactivation of the microtubule-destabilizing protein Op18 in the early steps of axon formation. Furthermore, after knockdown of DOCK7 in hippocampal neurons, axon formation was prevented, whereas overexpression of DOCK7 promoted the formation of multiple axons [2]. Like the other DOCK180 superfamily members, DOCK7 has two highly conserved domains: DHR-1 and DHR-2. The former is involved in phospholipid binding, and the latter is responsible for GEF catalytic activity [28]. The *DOCK7* frameshift variant identified in this study is located in the highly conserved DHR-2 domain and is predicted to abolish the functionality of this domain; thus, it is very likely to impede axon formation through Rac inactivation. Interestingly, results of an in vitro assay showed that lack of the DHR-2 domain failed to activate Rac1 [2]. Furthermore, the frameshift variant is located in exon 42 (out of 50 exons), and therefore most likely induces nonsense-mediated decay (NMD) of the corresponding mRNA.

The occurrence of UPDs is estimated as 1 in 3500 births, and UPDs have been described for nearly all chromosomes [25]. For the majority of chromosomes, UPD is without clinical consequences. However, some chromosomes (chromosomes 6, 7, 11, 14, 15, 20) are known to contain regions with parent-specific gene expression, and these can lead to clinically relevant consequences [22,29]. Additionally, UPDs can result in AR diseases due to homozygosity for a deleterious variant from a heterozygous parent. Several cases of AR disease caused by UPD events in chromosome 1 have been already reported [13,14,15,20,30,31]. No imprinting effects were observed in these cases. Based on the Geneimprint database, 18 candidate genes are likely to be imprinted (11 paternally, and 7 maternally expressed) in chromosome 1. However, no detailed studies have been published on imprinting effects related to chromosome 1 so far.

The majority of UPD cases arise from trisomic rescue and are associated with a meiosis I non-disjunction [32]. Whether non-disjunction occurred during meiosis I or II can be established by determining if the pericentromeric region is heterozygous (meiosis I) or homozygous (meiosis II). The mixed segmental maternal UPD identified in our patient can be explained by a meiosis II segregation error (non-disjunction) of chromosome 1 in maternal oogenesis due to a homozygous pericentromeric region [11]. Prior to meiotic segregation, four crossing over events must have occurred, and the recombinant and non-recombinant sister chromatids segregated into a disomic oocyte. After fertilization, the paternal chromosome 1 was eliminated by trisomic rescue, leading to mixed segmental maternal UPD. It is known that advanced maternal age can be a risk for segregation errors [32]. Our patient’s mother is 32 years old; therefore, her age is unlikely to explain the occurrence of the UPD. According to a genome-wide UPD prevalence study in the general population, UPD occurs most frequently in chromosomes 1, 4, 16, 21, 22, and X; UPD of chromosome 1 represents approximately 7% of all UPD cases, and maternal origin is three times higher than paternal origin. Due to the low prevalence of UPDs in the general population, the recurrence risk of an AR disease is rare.

Detecting UPD using genome-wide SNP arrays and WES is becoming a useful diagnostic approach to explain AR diseases with a discordant segregation (e.g., homozygosity for a potentially pathogenic single-nucleotide variant). However, UPD cases can still be missed when trio analysis followed by a SNP array analysis is not possible.

In conclusion, we identified a novel homozygous and potentially pathogenic frameshift variant (c.5391delA, p.(Ala1798Leu*fs*Ter59)) in the *DOCK7* gene arising from maternal UPiD in a female child patient. We suggest performing an SNP array analysis in such cases with discordant segregation, because one pathogenic variant can be transmitted from one parent and may result in homozygosity due to UPiD.

## 4. Materials and Methods

### 4.1. Patient and Family Members

This is a retrospective description of the ocular and general phenotype in a patient with early epileptic encephalopathy syndrome (EIEE). Informed written consent was obtained from the parents. This work adhered to the tenets of the Declaration of Helsinki. The affected patient received a detailed ophthalmological examination. On repeat visit, an electroretinogram (ERG) was performed according to the ISCEV standard [33] in mydriasis using a RETeval^®^ (LKC Technologies, Gaithersberg, MD, USA) handheld ERG device with the manufacturers’ proprietary Sensor Strip disposable skin electrodes. In addition, neurological examination, brain MRI, and standardized EEG recordings were performed.

### 4.2. Molecular Genetic Studies

Genomic DNA was extracted from EDTA-treated peripheral venous blood in duplicate with the automated chemagic MSM I system according to the manufacturer’s specifications (PerkinElmer Chemagen Technologie GmbH, Baesweiler, Germany) and stored at 4 °C.

#### 4.2.1. Whole-Exome Sequencing (WES) and Data Analysis

WES was conducted in the index patient and her parents. DNAs were fragmented using an M220 Sonicator (Covaris, Woburn, MA, USA), and ligation of adapters was performed according to the IDT-Illumina TruSeq DNA Exome protocol (Illumina Inc., San Diego, CA, USA). DNA fragments were then hybridized and enriched for coding regions with the xGEN Exome Research Panel v2.0 kit (Integrated DNA Technologies, Coralville, IA, USA) prepared according to the manufacturer’s protocol, followed by 75 bp pair-ended sequencing performed in house with the Illumina NextSeq 550 platform. FASTQ sequencing reads were mapped to the reference genome (GRCh37/hg19) with Burrows-Wheeler Aligner (BWA) Enrichment (Version 2.1.0, Illumina BaseSpace onsite) and variant calling with GATK (Illumina BaseSpace onsite). Performance metrics for each sample with the length of a targeted reference of 38′491′020 bp (coding and flanking intronic regions (±10 bp)) were: Proband (average sampling depth: 129.5×, target coverage at 10×: 98.7%), proband’s mother (average sampling depth: 118.6×, target coverage at 10×: 98.7%), and proband’s father (average sampling depth: 120.9×, target coverage at 10×: 98.9%). AlamutBatch version 1.11 (Sophia Genetics, Saint Sulpice, Switzerland) was used for annotating VCFs. Because the patient was initially suspected of having Leber congenital amaurosis, a comprehensive list of 509 genes of interest was used on WES data for filtering and analysis (available upon request). Genes associated with inherited retinal degenerations (IRDs) as well as candidate genes (selected from the relevant literature) were included in this list. A filtering pipeline was used to remove all frequent and benign variants. Variants with heterozygous allele frequency >1% (genome aggregation database (gnomAD) heterozygous allele frequency in all populations) and number of homozygous alternate alleles in gnomAD > 10 were excluded (https://gnomad.broadinstitute.org/ (accessed on 2 December 2021)). High priority was given to the variants if they met the following criteria: (1) loss-of-function (LoF) variants (nonsense, frameshift, canonical splicing) with <1% or unknown frequency in gnomAD, (2) classified as potentially disease-causing based on variant classification by the American College of Medical Genetics and Genomics (ACMG), (3) described as pathogenic in previous reports in Human Gene Mutation Database (HGMD), VarSome, or ClinVar. Copy number variants (CNVs) analysis was performed with the SeqNext module of the SeqPilot software 5.2.0 (JSI Medical Systems, Ettenheim, Germany).

#### 4.2.2. Single-Nucleotide Polymorphism (SNP) Array and Aneuploidy Screening (VeriSeq PGS)

Analysis using a single-nucleotide polymorphism (SNP) array (Illumina Infinium Karyomapping Assay, HumanKaryomap-12 DNA Analysis Kit) was performed on the index patient and her parents, while aneuploidy screening (Illumina VeriSeq PGS) was exclusively carried out with the proband’s DNA in order to investigate chromosomal abnormalities. The SNP array analysis was performed by using standardized protocols provided by the manufacturer. Images were captured on the iScan System of a NextSeq550 platform. Data were aligned to the human reference genome (hg19) and evaluated using the BlueFuse Multi 4.5 software (Illumina Inc., San Diego, CA, USA). Haplotyping was performed in order to determine the parental origin of the hetero- and isodisomic segmental regions in chromosome 1 by searching informative SNP markers within these regions. Analysis of LOH (Loss of Heterozygosity) was based on B-allele frequency calculation (BAF). The expected values for BAF are 0 for AA, 0.5 for AB, and 1 for BB. Copy number state was indicated by the Log R signal intensity ratio tract. Aneuploidy screening was performed using the VeriSeq PGS Kit (Illumina Inc., San Diego, CA, USA) according to the manufacturer’s guidelines, with a resolution of approximately 10 Mb.

#### 4.2.3. Analysis of Sequence Variants within the mUPD Regions of Chromosome 1

The WES data of the proband were filtered by selecting chromosome 1. Selected variants in chromosome 1 were then subdivided into both homozygous and heterozygous regions according to their genomic locations (hg19) displayed by the SNP array.

#### 4.2.4. Analysis of Intragenic Variants

Candidate sequence variants of interest and segregation analysis were confirmed by Sanger sequencing. PCR primer pairs were designed using the Primer3 program with genomic DNA sequences containing correspondent transcript sequences. Primer sequences and protocols are available upon request. Amplified PCR fragments were sequenced on a 16-capillary 3130*xl* Genetic Analyzer (Applied Biosystems, Waltham, MA, USA).

## Figures and Tables

**Figure 1 ijms-23-07382-f001:**
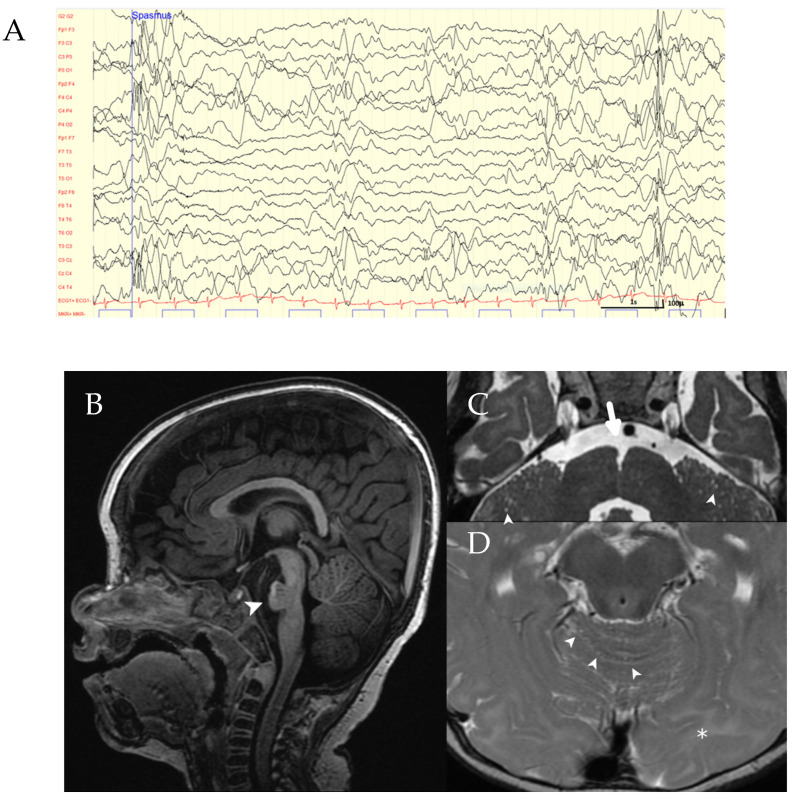
Clinical findings in the index patient. (**A**) EEG of the patient showing generalized epileptiform discharges. (**B**–**D**) Brain MRI: Sagittal T1-weighted image showing pontine hypoplasia (arrowhead in (**B**)). Axial T2-weighted images (**C**,**D**) demonstrating an abnormally marked anterior median fissure at the pontomedullary junction (large arrow in (**C**)), multiple microcysts in the cerebellar hemispheres and cerebellar vermis (arrowheads in (**C**,**D**)), and hyperintense signal of the occipital white matter (asterisk in (**D**)).

**Figure 2 ijms-23-07382-f002:**
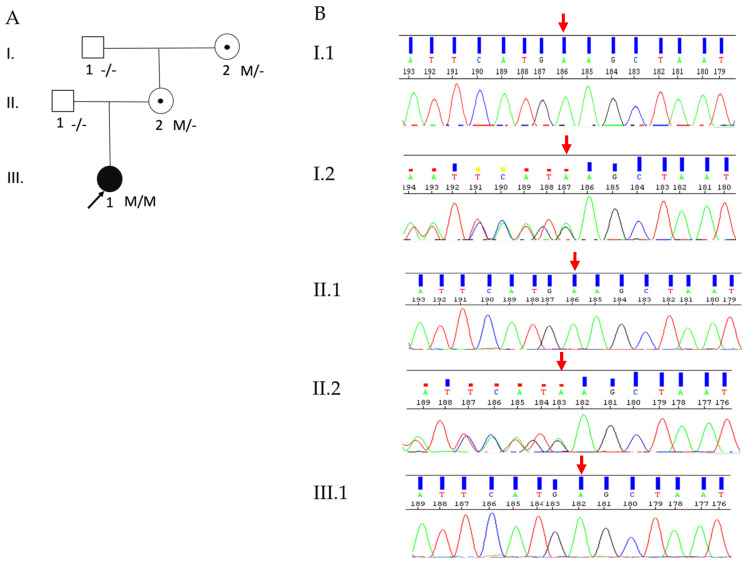
Pedigree and Sanger sequencing profiles: (**A**) Pedigree of the index patient. The black arrow represents the patient. Symbols with central dots indicate heterozygous carriers. Genotypes are indicated below the pedigree symbols. M represents the sequence variant c.5391delA (p.(Ala1798LeufsTer59)) in the DOCK7 gene, and (-) indicates the reference allele. (**B**) Shown are the electropherograms of five family members: the homozygous sequence variant c.5391delA (p.(Ala1798LeufsTer59)) was identified in the index patient (III.1). Her mother (II.2) and the maternal grandmother (I.2) were heterozygous, while her father (II.1) and maternal grandfather (I.1) were non-carriers for this variant. The red arrow indicates the variant position.

**Figure 3 ijms-23-07382-f003:**
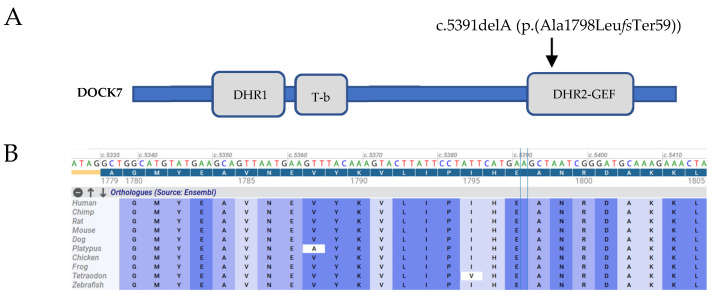
(**A**) Schematic of DOCK7 showing the relative location of the frameshift variant c.5391delA in Exon 42 (NM_001271999.1). DOCK homology domains DHR1 (amino acids 516–727) and DHR2 (amino acids 1668–2110) and TACC3-binding region (T-b). (**B**) Evolutionary conservation of the frameshift variant among 10 vertebrates.

**Figure 4 ijms-23-07382-f004:**
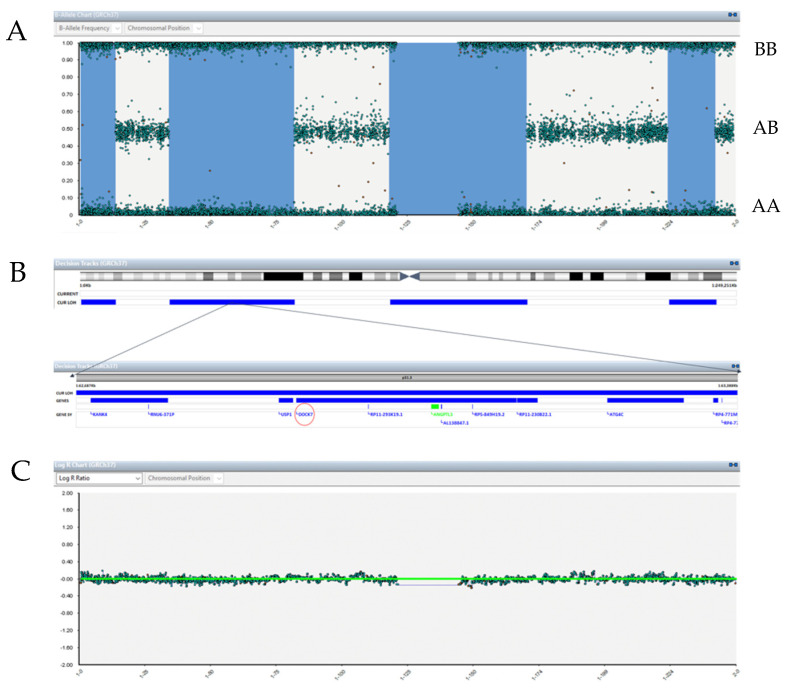
Details of maternal UPD by SNP array analysis of chromosome 1. (**A**) B-allele chart displayed four segments of chromosome 1 alternately homozygous (UPiD; shown in blue, no heterozygous SNPs, only AA and BB)) and heterozygous (UPhD; SNPs with AB genotype present) regions. (**B**) The *DOCK7* gene (red circle) is located in p31.3, a region situated in one of the four homozygous segments. (**C**) Aneuploidy screening (Log R Ratio) did not show aneuploidies or CNVs >10 Mb.

## Data Availability

Not applicable.

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
