# Peer review of "Homozygosity for a Novel DOCK7 Variant Due to Segmental Uniparental Isodisomy of Chromosome 1 Associated with Early Infantile Epileptic Encephalopathy (EIEE) and Cortical Visual Impairment"

_ijms, 2022, doi:10.3390/ijms23137382_

Round 1

Reviewer 1 Report

This is a well written paper and I only have two minor comments. The first is that ClinGen is updating guidance for use of ACMG/AMP pathogenicity criteria and their current guidance should be followed when reporting on pathogenicity classifications of variants. You can read the guidance at https://clinicalgenome.org/working-groups/sequence-variant-interpretation/, however, I have two suggestions. PM2 is now suggested to be used at supporting not at moderate and PP3 should not be used in conjunction with PVS1 (PMID: 30192042). The variant will still classify as pathogenic. My other suggestion is to include two additional variants found in the following publications (from HGMD), PMID:32552793 and PMID:31054490.

Reviewer 2 Report

Major issues

As a case report, this is a well written paper. However, EIEE aka Ohtahara syndrome is well known that has gene mutation. Even though this gene is the first case, I do not think that this case report significantly contributes to the medical field.

Minor issue

MRI my be okay to use, but EEG etc. should be clearly written at first use.

Author Response

Please see attachement.

Round 2

Reviewer 2 Report

 As I have already commented that a case that showed rareness is not good enough.